# Forced Zika Virus Infection of *Culex pipiens* Leads to Limited Virus Accumulation in Mosquito Saliva

**DOI:** 10.3390/v12060659

**Published:** 2020-06-19

**Authors:** Sandra R. Abbo, Chantal B. F. Vogels, Tessa M. Visser, Corinne Geertsema, Monique M. van Oers, Constantianus J. M. Koenraadt, Gorben P. Pijlman

**Affiliations:** 1Laboratory of Virology, Wageningen University & Research, Droevendaalsesteeg 1, 6708 PB Wageningen, The Netherlands; sandra.abbo@wur.nl (S.R.A.); corinne.geertsema@wur.nl (C.G.); monique.vanoers@wur.nl (M.M.v.O.); 2Laboratory of Entomology, Wageningen University & Research, Droevendaalsesteeg 1, 6708 PB Wageningen, The Netherlands; chantal.vogels@yale.edu (C.B.F.V.); tessa.visser@wur.nl (T.M.V.); sander.koenraadt@wur.nl (C.J.M.K.)

**Keywords:** Zika virus, flavivirus, arbovirus, *Culex pipiens*, vector competence, midgut barrier, the Netherlands

## Abstract

Zika virus (ZIKV) is a mosquito-borne pathogen that caused a large outbreak in the Americas in 2015 and 2016. The virus is currently present in tropical areas around the globe and can cause severe disease in humans, including Guillain-Barré syndrome and congenital microcephaly. The tropical yellow fever mosquito, *Aedes aegypti*, is the main vector in the urban transmission cycles of ZIKV. The discovery of ZIKV in wild-caught *Culex* mosquitoes and the ability of *Culex quinquefasciatus* mosquitoes to transmit ZIKV in the laboratory raised the question of whether the common house mosquito *Culex pipiens*, which is abundantly present in temperate regions in North America, Asia and Europe, could also be involved in ZIKV transmission. In this study, we investigated the vector competence of *Cx. pipiens* (biotypes *molestus* and *pipiens*) from the Netherlands for ZIKV, using Usutu virus as a control. After an infectious blood meal containing ZIKV, none of the tested mosquitoes accumulated ZIKV in the saliva, although 2% of the *Cx. pipiens pipiens* mosquitoes showed ZIKV–positive bodies. To test the barrier function of the mosquito midgut on virus transmission, ZIKV was forced into *Cx. pipiens* mosquitoes by intrathoracic injection, resulting in 74% (*molestus*) and 78% (*pipiens*) ZIKV–positive bodies. Strikingly, 14% (*molestus*) and 7% (*pipiens*) of the tested mosquitoes accumulated ZIKV in the saliva after injection. This is the first demonstration of ZIKV accumulation in the saliva of *Cx. pipiens* upon forced infection. Nevertheless, a strong midgut barrier restricted virus dissemination in the mosquito after oral exposure and we, therefore, consider *Cx. pipiens* as a highly inefficient vector for ZIKV.

## 1. Introduction

Mosquito-borne viruses are a severe threat to human health [1,2]. Climate change, increased global trade and travel, and the ability of viruses to adapt to new vectors and hosts contribute to the geographic expansion of these mosquito-borne pathogens [1,2]. Zika virus (ZIKV; family *Flaviviridae*, genus *Flavivirus*) was first isolated from a caged, sentinel rhesus monkey in the canopy of the Zika forest in Uganda in 1947 [2,3]. In 1948, the virus was discovered in *Aedes africanus* mosquitoes in the same forest [2,3], and in 1954 the first human ZIKV isolate was obtained from a Nigerian female [2,4]. Much later, the virus re-emerged in Asia and the Pacific islands, and started a large outbreak in humans in Brazil in 2015 [2,5,6]. Historically, ZIKV infection results in a mild, self-limiting febrile illness for an estimated 20% of the infected individuals [7]. However, during the outbreak in the Americas, ZIKV infections in humans unexpectedly caused severe diseases, such as Guillain–Barré syndrome and congenital Zika syndrome including microcephaly [7,8].

The widespread distribution of the yellow fever mosquito, *Aedes aegypti*, and the Asian tiger mosquito, *Aedes albopictus*, in Central and South America [9] may have favored the rapid emergence of ZIKV across the Western Hemisphere. Field studies and laboratory vector competence experiments have shown that mosquitoes of the *Aedes* genus are the main vectors in both the sylvatic and urban transmission cycles of ZIKV [3,10,11,12,13,14,15,16]. However, the discovery of ZIKV in field-collected *Culex* mosquitoes [17,18,19,20] and the recent demonstration of experimental ZIKV transmission by *Culex quinquefasciatus* [21,22] have posed the question of whether the common house mosquito, *Culex pipiens*, could also be involved in ZIKV transmission. Since *Cx. pipiens* mosquitoes are abundantly present in temperate regions in North America, Asia and Europe [23,24], where the ZIKV vectors *Ae. aegypti* and *Ae. albopictus* are less dominant [9,25], ZIKV transmission by both *Aedes* and *Culex* vectors would greatly increase the human population size at risk for ZIKV infection.

*Cx. pipiens* can be found in two morphologically indistinguishable biotypes, *pipiens* and *molestus*, which differ in behavior, physiology and genetic background. The *pipiens* biotype prefers to feed on birds, diapauses during winter and requires a blood meal to lay eggs [26]. The *molestus* biotype prefers to bite mammals, including humans, remains active during winter and does not need a blood meal to lay the first batch of eggs [26]. *Cx. pipiens* is a competent vector for the flaviviruses West Nile virus (WNV) and Usutu virus (USUV) [24,27,28,29]. So far, *Cx. pipiens* has shown to be an incompetent vector for ZIKV during vector competence experiments [30,31,32,33,34,35,36], although important positive controls for the competence of the tested mosquitoes and the infectivity of the viruses have not always been included.

The aim of this study was to determine the vector competence of *Cx. pipiens* (biotypes *molestus* and *pipiens*) from the Netherlands for ZIKV. We investigated whether or not *Cx. pipiens* mosquitoes could experimentally transmit ZIKV after an infectious blood meal. We tested high numbers of mosquitoes, and we infected *Cx. pipiens* with USUV and *Ae. aegypti* with ZIKV as positive controls. We also injected ZIKV into the thoraxes of *Cx. pipiens* mosquitoes to study the viral replication dynamics and the barrier function of the mosquito midgut on ZIKV transmission.

## 2. Materials and Methods

### 2.1. Mosquito Rearing

Colonies of *Cx. pipiens molestus* and *Cx. pipiens pipiens* from the Netherlands [29] were maintained at 23 °C with 60% relative humidity and a 16:8 light:dark period. Mosquitoes were reared as previously described [29]. Egg rafts were placed in trays with tap water and Liquifry No. 1 (Interpet Ltd., Dorking, UK). Emerged larvae were fed daily with TetraMin baby fish food (Tetra, Melle, Germany). Pupae were allowed to emerge in 30 cm cubic Bugdorm cages, and the adults were provided with a 6% glucose solution. 

*Ae. aegypti* mosquitoes (Rockefeller strain, obtained from Bayer AG, Monheim, Germany) were maintained as described earlier [37]. Mosquitoes were reared at 27 °C with 70% relative humidity and a 12:12 light:dark period. Adult female mosquitoes laid their eggs on moist filter paper that was placed in a cup containing tap water. The eggs were air-dried for 3–4 days and then placed in trays containing tap water with Liquifry No. 1 (Interpet Ltd.). The larvae were fed with TetraMin baby fish food (Tetra). Adults were kept in 30 cm cubic Bugdorm cages with access to a 6% glucose solution. 

### 2.2. Cells and Viruses

African green monkey kidney Vero E6 cells were grown as a monolayer in Dulbecco’s Modified Eagle Medium (DMEM; Gibco, Carlsbad, CA, USA) supplemented with 10% fetal bovine serum (FBS; Gibco), penicillin (100 U/mL; Sigma-Aldrich, Saint Louis, MO, USA) and streptomycin (100 μg/mL; Sigma-Aldrich) (P/S). The cells were maintained at 37 °C with 5% CO_2_. Prior to virus infections, the Vero cells were seeded in HEPES-buffered DMEM medium (Gibco) supplemented with 10% FBS and P/S. When mosquito body lysate or saliva was added to the cells, the HEPES-buffered DMEM medium was additionally supplemented with gentamycin (50 μg/mL; Gibco) and fungizone (2.5 μg/mL of amphotericin B and 2.1 μg/mL of sodium deoxycholate; Gibco). This medium will hereafter be named DMEM HEPES complete. 

All experiments involving infectious ZIKV and USUV were executed in the biosafety level 3 laboratory of Wageningen University & Research. Passage 5 and 6 virus stocks of ZIKV, Suriname 2016 (GenBank accession no. KU937936.1; EVAg Ref-SKU 011V-01621; obtained from Erasmus Medical Center, Rotterdam, the Netherlands), and passage 6 virus stocks of USUV, the Netherlands 2016 (GenBank accession no. MH891847.1; EVAg Ref-SKU 011V-02153; obtained from Erasmus Medical Center), were grown on Vero cells. Viral titers, expressed as 50% tissue culture infectious dose per milliliter (TCID_50_/mL), were measured using end point dilution assays (EPDAs) on Vero cells in 60-well MicroWell plates (Nunc, Roskilde, Denmark). 

### 2.3. Infectious Blood Meal

Prior to the infectious blood meal, female mosquitoes were starved for one day. Mosquitoes were then orally exposed to ZIKV or USUV by providing them with infectious blood from a Hemotek PS5 feeder (Discovery Workshops, Lancashire, UK) in a dark room for 1 h. Infectious blood meals were prepared by mixing virus stock with human blood (Sanquin Blood Supply Foundation, Nijmegen, the Netherlands) to obtain a final virus titer of 1.0 × 10^7^ TCID_50_/mL. *Cx. pipiens molestus* and *Cx. pipiens pipiens* received ZIKV during four and three independent experiments, respectively. As positive controls, *Ae. aegypti* was infected with ZIKV to test the quality of the virus stock, and *Cx. pipiens molestus* and *Cx. pipiens pipiens* were exposed to USUV to test the competence of the mosquitoes. After the blood meal, the mosquitoes were anesthetized using CO_2_, and fully engorged females were selected. A small number of engorged females was collected in individual SafeSeal micro tubes (Sarstedt, Nümbrecht, Germany) containing 0.5 mm zirconium oxide beads (Next Advance, Averill Park, NY, USA) to determine the virus titer in the mosquito body directly after engorgement. All other females were incubated at 28 °C with access to 6% glucose.

### 2.4. Intrathoracic Injection

Female mosquitoes were immobilized with CO_2_ prior to intrathoracic injection using a Drummond Nanoject II Auto-Nanoliter Injector (Drummond Scientific, Broomall, PA, USA). *Cx. pipiens molestus* was injected with 1.0 × 10^4^ TCID_50_ of ZIKV or 3.5 × 10^3^ TCID_50_ of USUV (positive control). *Cx. pipiens pipiens* was injected with decreasing doses of ZIKV containing 1.0 × 10^4^, 1.0 × 10^2^ or 3.0 × 10^1^ TCID_50_, or with 3.5 × 10^3^ TCID_50_ of USUV (positive control). *Ae. aegypti* was injected with 1.0 × 10^4^ TCID_50_ of ZIKV (positive control). The injected mosquitoes were incubated at 28 °C with access to 6% glucose.

### 2.5. Salivation Assay

Fourteen days post infection, the mosquitoes were immobilized with CO_2_ and the legs and wings of each mosquito were removed. Next, mosquito saliva was collected by inserting the mosquito proboscis into a 200 µL pipet tip containing 5 µL of a 50% FBS and 25% sugar solution in sterilized tap water. After 45 min, the mosquito bodies were stored at −80 °C in individual SafeSeal micro tubes (Sarstedt) containing 0.5 mm zirconium oxide beads (Next Advance). Individual mosquito saliva samples were mixed with 55 µL DMEM HEPES complete and stored at −80 °C. 

### 2.6. Infectivity Assay

Frozen mosquito body samples were homogenized in a Bullet Blender Storm (Next Advance) at maximum speed for 2 min. The body homogenates were centrifuged in an Eppendorf 5424 centrifuge at 14,500 rpm for 1 min. Next, 100 µL of DMEM HEPES complete was added to each body homogenate. The homogenates in medium were blended again at maximum speed for 2 min, and centrifuged at 14,500 rpm for 2 min. Thirty µL of each mosquito body or saliva sample was then added to one well of a 96-well plate containing a monolayer of Vero cells in DMEM HEPES complete. After 2 h at 37 °C, the medium of the cells was replaced by 100 µL fresh DMEM HEPES complete. Six days post infection, the wells were scored virus–positive or –negative based on cytopathic effect (CPE). The number of virus–positive mosquito bodies or salivas was expressed as a percentage of the total number of mosquitoes tested. Viral titers in TCID_50_/mL were measured for mosquito bodies and salivas using EPDAs on Vero cells. After 6 days, the wells were scored virus–positive or –negative based on CPE. 

### 2.7. RNA Extraction and Reverse Transcriptase PCR

Total RNA was isolated from Vero cells using TRIzol reagent (Invitrogen, Carlsbad, CA, USA) according to the manufacturer’s protocol. RNA yields were measured using a NanoDrop ND-1000 spectrophotometer. Reverse transcriptase PCR (RT-PCR) was done using a 2720 Thermal Cycler (Applied Biosystems, Foster City, CA, USA) and the SuperScript III One-Step RT-PCR System with Platinum *Taq* DNA polymerase (Invitrogen), according to the manufacturer’s protocol. Per RT-PCR reaction, 100 ng of total RNA was added. Primers targeting the region encoding ZIKV non-structural protein 1 (NS1) (forward: 5′-GAGACGAGATGCGGTACAGG-3′; reverse: 5′-CGACCGTCAGTTGAACTCCA-3′) and the region coding for USUV non-structural protein 5 (NS5) (forward: 5′-GGCTGTAGAGGACCCTCGG-3′; reverse: 5′-GACTGCCTTTCGCTTTGCCA-3′) were used at annealing temperatures of 55 °C and 60 °C, respectively.

### 2.8. Mosquito Wing Length Measurement

The right wings of 20 female *Cx. pipiens molestus*, *Cx. pipiens pipiens* and *Ae. aegypti* were removed and mounted on sticky tape on a slide. The wing length was measured from the end of the alula to the top of the wing, excluding the fringe scales, using ImageFocus software (Euromex Microscopes, Arnhem, the Netherlands) calibrated with a slide graticule of 0.01 mm. The wing length measurements were used as an estimate of body size, as wing length is known to be correlated with body mass [38].

### 2.9. Statistical Analysis

The Kolmogorov–Smirnov test was used to determine whether mosquito wing lengths and viral titers of engorged mosquitoes were normally distributed. Differences in wing lengths and differences in viral titers were then tested for significance using an unpaired, two-tailed *t*-test. All statistical tests were performed using GraphPad Prism 5 (GraphPad Software, San Diego, CA, USA). 

## 3. Results

### 3.1. No ZIKV Transmission by Cx. pipiens after an Infectious Blood Meal

To assess the vector competence of *Cx. pipiens molestus* and *Cx. pipiens pipiens* for ZIKV, the mosquitoes were offered an infectious blood meal containing 1.0 × 10^7^ TCID_50_/mL of ZIKV. As positive controls, *Ae. aegypti* and both *Cx. pipiens* biotypes were infected with 1.0 × 10^7^ TCID_50_/mL of ZIKV or USUV, respectively. To investigate the variability in engorgement among individual mosquitoes, viral body titers were determined for a selection of mosquitoes directly after ingestion of an infectious blood meal (Figure 1A,B). *Cx. pipiens* mosquitoes, blood fed with ZIKV or USUV, showed very similar median viral titers ranging from 4.6 × 10^5^ to 6.3 × 10^5^ TCID_50_/mL. *Ae. aegypti* mosquitoes blood fed with ZIKV showed significantly lower viral titers compared to the *Cx. pipiens* biotypes blood fed with ZIKV (*p* < 0.05). This can be explained by the fact that the *Ae. aegypti* mosquitoes were smaller in size than the *Cx. pipiens* mosquitoes. The *Ae. aegypti* mosquitoes showed an average wing length (±standard deviation) of 2.60 mm (±0.18 mm), whereas average wing lengths of 3.34 mm (±0.32 mm) and 3.65 mm (±0.24 mm) were measured for *molestus* and *pipiens*, respectively. The measured wing lengths of *Ae. aegypti* were significantly lower compared to the wing lengths of each *Cx. pipiens* biotype (*p* < 0.0001). The smaller size of *Ae. aegypti* mosquitoes likely results in a lower volume of ingested blood containing virus.

After the infectious blood meal, all other engorged mosquitoes were incubated at 28 °C for 14 days, and afterwards the mosquito bodies and salivas were tested for the presence of virus using infectivity assays on Vero cells. The presence of virus was scored based on CPE, and the presence of viral RNA was also confirmed using RT-PCR for a subset of the results. None of the 55 tested *Cx. pipiens molestus* mosquitoes showed a ZIKV–positive body or saliva (Figure 2A,B). Out of the 133 *Cx. pipiens pipiens* tested, two mosquitoes showed a ZIKV–positive body but no positive saliva (Figure 2A,B). For ZIKV-blood fed *Ae. aegypti* mosquitoes, which served as positive controls, 100% of the tested mosquitoes were infected and 65% showed virus–positive saliva (Figure 2A,B), which corresponds with our previous work [37]. In addition, the USUV-blood fed *Cx. pipiens* showed 67% (*molestus*) and 88% (*pipiens*) virus–positive bodies (Figure 2A), and 31% (*molestus*) and 21% (*pipiens*) virus–positive salivas (Figure 2B), demonstrating that both *Cx. pipiens* biotypes were competent vectors for USUV. Given that none of the *Cx. pipiens* mosquitoes accumulated ZIKV in the saliva after oral infection, we conclude that *Cx. pipiens* is a highly inefficient vector for ZIKV.

### 3.2. Low ZIKV Titers in Cx. pipiens after an Infectious Blood Meal

Viral titers of virus–positive bodies and salivas were measured by EPDAs. The two ZIKV–positive *Cx. pipiens pipiens* bodies both had a viral titer of 6.3 × 10^3^ TCID_50_/mL (Figure 3A), whereas the viral body titers of *Ae. aegypti* mosquitoes were very high with a median titer of 1.1 × 10^7^ TCID_50_/mL (Figure 3B). The median viral saliva titer of *Ae. aegypti* was below the detection limit of 1.0 × 10^3^ TCID_50_/mL (Figure 3B). For USUV-blood fed *Cx. pipiens*, the median viral body titers were 8.0 × 10^4^ TCID_50_/mL (*molestus*) and 2.9 × 10^5^ TCID_50_/mL (*pipiens)*, and the median viral saliva titers were below the detection limit of 1.0 × 10^3^ TCID_50_/mL (*molestus*) and 6.3 × 10^3^ TCID_50_/mL (*pipiens*) (Figure 3C,D). These results showed that ZIKV had the ability to infect *Cx. pipiens* and replicate in the mosquito with very low efficiency. 

### 3.3. Intrathoracic ZIKV Injection Leads to Virus Replication in Cx. pipiens with Limited Dissemination to the Mosquito Saliva

To investigate whether ZIKV was unable to pass the mosquito midgut barrier in *Cx. pipiens*, mosquitoes were injected in the thorax with 1.0 × 10^4^ TCID_50_ of ZIKV. As positive controls, *Ae. aegypti* was injected with ZIKV and both *Cx. pipiens* biotypes were injected with USUV. After 14 days at 28 °C, mosquito bodies and salivas were analyzed for the presence of virus. During infectivity assays, the presence of virus was scored based on CPE and, for a subset of the results, these scores were also confirmed by RT-PCR. High percentages (74% for *molestus* and 78% for *pipiens*) of the injected mosquitoes had ZIKV-positive bodies (Figure 4A). Interestingly, 14% (*molestus*) and 7% (*pipiens*) of the injected mosquitoes also showed infectious ZIKV in their saliva (Figure 4B). After injection of *Ae. aegypti* with ZIKV, which served as a positive control experiment, 100% of the injected mosquitoes showed virus–positive bodies (Figure 4A), whereas 72% of the injected mosquitoes showed virus–positive saliva (Figure 4B), which is in line with our earlier results [37]. As another positive control, both biotypes of *Cx. pipiens* were injected with USUV, which showed that 100% of the tested *Cx. pipiens molestus* and *Cx. pipiens pipiens* had USUV–positive bodies (Figure 4A), and 94% (*molestus*) and 88% (*pipiens*) of the injected mosquitoes had USUV–positive saliva (Figure 4B). This indicates that virus dissemination to the saliva in *Cx. pipiens* is more efficient with USUV than with ZIKV.

### 3.4. Effect of Injected Viral Dose on ZIKV Infection of Cx. pipiens

To investigate whether the percentage of ZIKV–positive mosquitoes after intrathoracic injection was affected by the viral dose provided, *Cx. pipiens pipiens* mosquitoes were also injected with lower doses of ZIKV. After 14 days, 40% of the *Cx. pipiens pipiens* mosquitoes injected with 1.0 × 10^2^ TCID_50_ of ZIKV showed virus–positive bodies, whereas 2% showed virus–positive salivas (Figure 4A,B). Additionally, when a viral dose of 3.0 × 10^1^ TCID_50_ was supplied, 13% of the tested *Cx. pipiens pipiens* mosquitoes showed virus–positive bodies, whereas none of the mosquitoes showed virus–positive saliva (Figure 4A,B). This indicated that the infection and transmission potential of ZIKV-injected *Cx. pipiens pipiens* was dependent on the viral dose provided, but also that a low ZIKV dose of 3.0 × 10^1^ TCID_50_ can infect a mosquito. 

### 3.5. Variability of Viral Titers in ZIKV-Injected Cx. pipiens

To obtain better insight into the ZIKV replication dynamics in injected *Cx. pipiens* mosquitoes, viral body and saliva titers were measured by EPDAs. To validate ZIKV replication in the primary ZIKV vector *Ae. aegypti* (positive control), the bodies and salivas of ZIKV-injected *Ae. aegypti* were also titrated. ZIKV body titers in *Cx. pipiens* injected with a dose of 1.0 × 10^4^ TCID_50_ were highly variable with maximum titers of 1.1 × 10^6^ TCID_50_/mL (*molestus*) and 6.3 × 10^5^ TCID_50_/mL (*pipiens*) (Figure 5A,B). This showed that ZIKV is intrinsically capable of replication to high viral titers in *Cx. pipiens*. The median viral body titers of ZIKV-injected *Cx. pipiens* were 9.6 × 10^3^ TCID_50_/mL for *pipiens* and below the detection limit of 1.0 × 10^3^ TCID_50_/mL for *molestus*. For *Ae. aegypti*, a high median viral body titer of 6.32 × 10^6^ TCID_50_/mL was found (Figure 5C). 

Interestingly, four *Cx. pipiens molestus* mosquitoes with ZIKV–positive saliva showed viral body titers below the detection limit of 1.0 × 10^3^ TCID_50_/mL (Figure 5A). Moreover, out of the three ZIKV-injected *Cx. pipiens pipiens* mosquitoes with ZIKV–positive saliva, two mosquitoes had relatively high viral body titers of 3.6 × 10^5^ TCID_50_/mL and 5.0 × 10^5^ TCID_50_/mL, whereas the third mosquito had a relatively low viral body titer of 8.0 × 10^3^ TCID_50_/mL (Figure 5B). This indicates that viral dissemination into the saliva of *Cx. pipiens* does not always correlate with a high viral titer in the mosquito body. 

## 4. Discussion

In this study, we set out to determine the vector competence of Dutch *Cx. pipiens* (biotypes *molestus* and *pipiens*) for ZIKV and we found that *Cx. pipiens* was unable to experimentally transmit ZIKV after an infectious blood meal. However, to the best of our knowledge, our study is the first study to demonstrate ZIKV dissemination to the saliva of *Cx. pipiens* mosquitoes after intrathoracic injection. Previous studies reported the presence of ZIKV in *Cx. pipiens* bodies [30,31] and heads [30] after injection but no virus accumulation in the saliva [30,31]. This suggests that ZIKV is incapable of infecting the salivary glands and/or of entering the saliva of *Cx. pipiens* [31]. We showed, however, that ZIKV can accumulate in the mosquito saliva after an intrathoracic injection with a viral dose as low as 1.0 × 10^2^ TCID_50_. This viral dose for injection is similar or lower compared to other studies that did not report ZIKV presence in the saliva [30,31]. Based on our results, we conclude that ZIKV is intrinsically capable of dissemination to the saliva of *Cx. pipiens* upon forced infection.

Nonetheless, even after forced infection, ZIKV replication in *Cx. pipiens* appeared to be suboptimal, as 74% (*molestus*) and 78% (*pipiens*) of the *Cx. pipiens* injected with a viral dose of 1.0 × 10^4^ TCID_50_ of ZIKV showed virus–positive bodies compared to 100% of the ZIKV-injected *Ae. aegypti* and USUV-injected *Cx. pipiens*. These results suggest a general replication deficiency of ZIKV in *Culex* cells, which has also been observed by others [31]. The underlying mechanisms responsible for the specific restriction of ZIKV in *Cx. pipiens* are currently unknown and need further investigation. Important factors that should be considered are physical barriers at the mosquito midgut and salivary glands, mosquito host factors required for virus replication, mosquito immune responses and the mosquito midgut microbiome [39]. Flaviviruses, such as ZIKV, but also yellow fever virus and dengue virus, are primarily associated with *Aedes* vectors [40,41]. Other flaviviruses, such as WNV and USUV, are mainly associated with *Culex* vectors [40,41]. Genetic differences between the mosquito genera and/or between the respective flaviviruses likely constrain and maintain the observed vector specificity. However, arboviruses have previously been shown to have the potential to quickly adapt to new vectors [42,43] and, therefore, it is important to investigate the molecular basis underlying the vector specificity of ZIKV, and also to investigate whether virus evolutionary trajectories can be predicted that could potentially lead to new epidemic variants of ZIKV with altered vector specificity.

We provided evidence via injection experiments that the mosquito midgut acted as an important barrier against ZIKV dissemination in *Cx. pipiens*. Our finding, with regard to the inability of *Cx. pipiens* to transmit ZIKV, is in line with other recent studies suggesting that mosquitoes of the *Culex* genus are poor ZIKV vectors [30,31,32,33,34,35,36,44]. Even at an incubation temperature as high as 28 °C, *Cx. pipiens* did not accumulate ZIKV in the saliva after oral exposure [30,31,34]. However, when considering the massive number of mosquitoes in the field and the fact that our laboratory study only measures vector competence and does not take into account all factors contributing to the vectorial capacity of the *Cx. pipiens* mosquito species [39], we cannot completely rule out the possibility of ZIKV transmission by *Cx. pipiens* in the field. Nevertheless, the reports of others [30,31,32,33,34,35,36] and our experiments with high numbers of tested *Cx. pipiens* mosquitoes and positive controls to validate the competence of the tested *Cx. pipiens* colonies, and the used ZIKV isolate, consolidate the conclusion that *Cx. pipiens* is a highly inefficient vector for ZIKV.

## Figures and Tables

**Figure 1 viruses-12-00659-f001:**
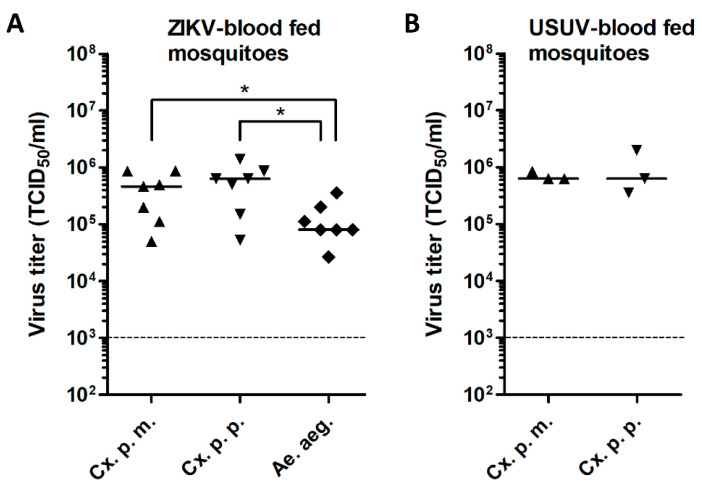
Virus titers in engorged *Cx. pipiens molestus* (Cx. p. m.), *Cx. pipiens pipiens* (Cx. p. p.), and *Ae. aegypti* (Ae. aeg.) mosquitoes immediately after ingestion of a blood meal containing (**A**) ZIKV or (**B**) USUV. Data points show individual mosquitoes exposed to ZIKV or USUV. Lines show median virus titers. Dashed lines indicate the detection limit of the EPDA. Asterisks indicate a significant difference (*p* < 0.05, *t*-test).

**Figure 2 viruses-12-00659-f002:**
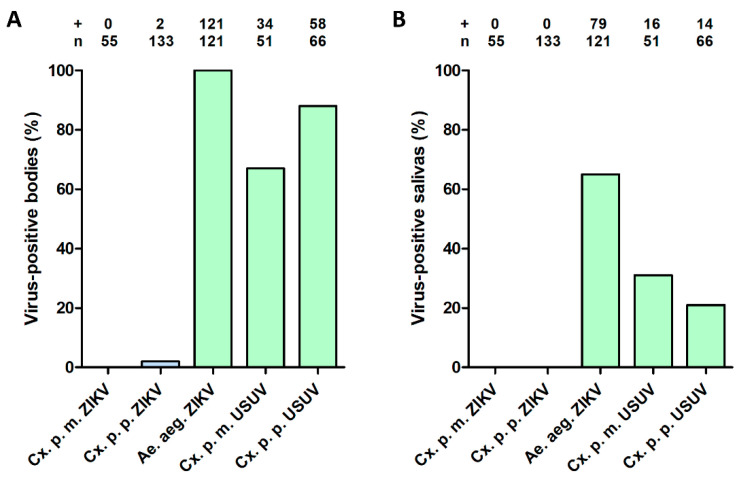
(**A**) Infection and (**B**) transmission of ZIKV and USUV after oral exposure to *Cx. pipiens molestus* (Cx. p. m.)*, Cx. pipiens pipiens* (Cx. p. p.) and *Ae. aegypti* (Ae. aeg.). After the infectious blood meal, mosquitoes were incubated at 28 °C for 14 days. The number of virus–positive mosquito bodies or salivas (indicated by +) is expressed as a percentage of the total number of mosquitoes tested (indicated by *n*). Experimental groups are depicted in blue; positive controls are depicted in green.

**Figure 3 viruses-12-00659-f003:**
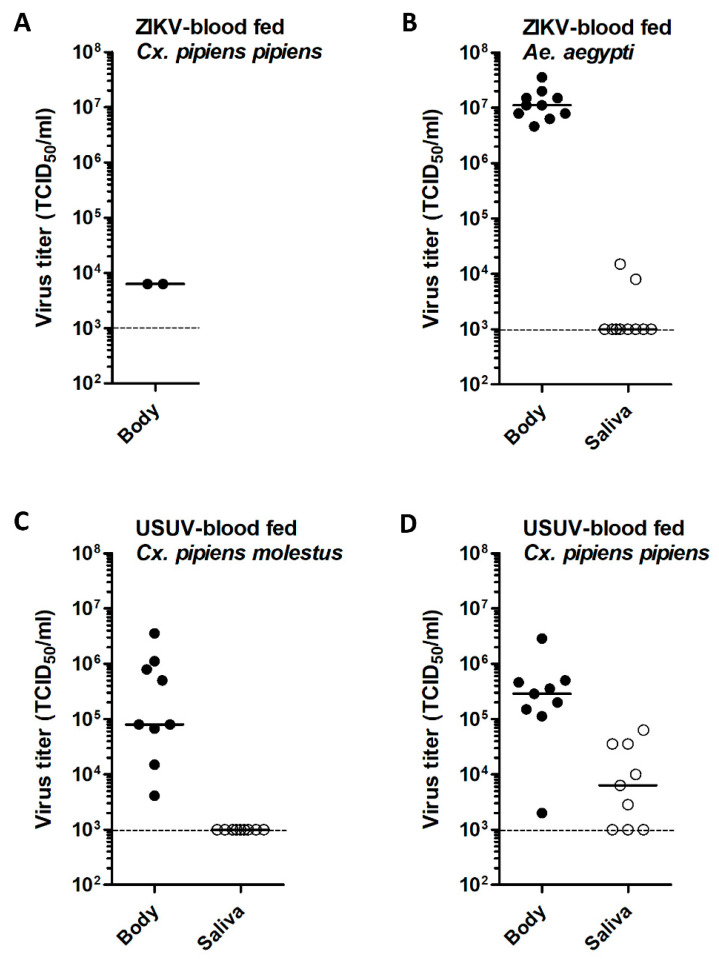
Virus titers in bodies and salivas of *Cx. pipiens molestus*, *Cx. pipiens pipiens*, and *Ae. aegypti* mosquitoes after oral exposure to ZIKV or USUV. After oral exposure, mosquitoes were incubated at 28 °C for 14 days. Virus titers were determined by EPDA for (**A**) bodies of ZIKV-blood fed *Cx. pipiens pipiens*, (**B**) bodies and salivas of ZIKV-blood fed *Ae. aegypti*, (**C**) bodies and salivas of USUV-blood fed *Cx. pipiens molestus*, (**D**) bodies and salivas of USUV-blood fed *Cx. pipiens pipiens*. Data points show individual mosquitoes infected with ZIKV or USUV. Lines show median virus titers. Dashed lines indicate the detection limit of the EPDA.

**Figure 4 viruses-12-00659-f004:**
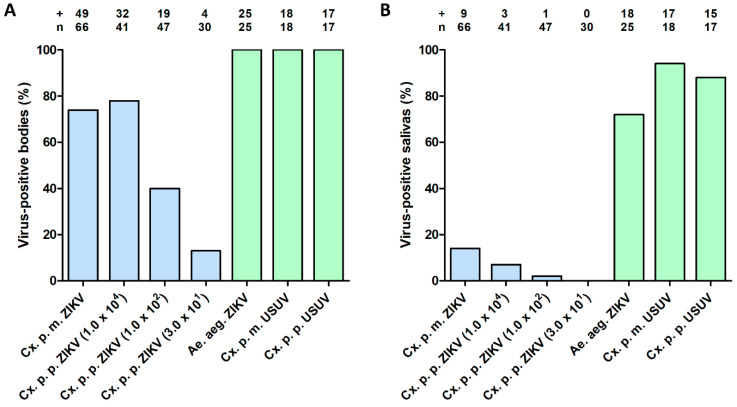
(**A**) Infection and (**B**) transmission of ZIKV and USUV after intrathoracic injection in *Cx. pipiens molestus* (Cx. p. m.), *Cx. pipiens pipiens* (Cx. p. p.) and *Ae. aegypti* (Ae. aeg.). *Cx. pipiens pipiens* mosquitoes were injected with three different ZIKV doses, as indicated (in TCID_50_). The injected mosquitoes were incubated at 28 °C for 14 days. The number of virus–positive mosquito bodies or salivas (indicated by +) is expressed as a percentage of the total number of mosquitoes tested (indicated by *n*). Experimental groups are depicted in blue; positive controls are depicted in green.

**Figure 5 viruses-12-00659-f005:**
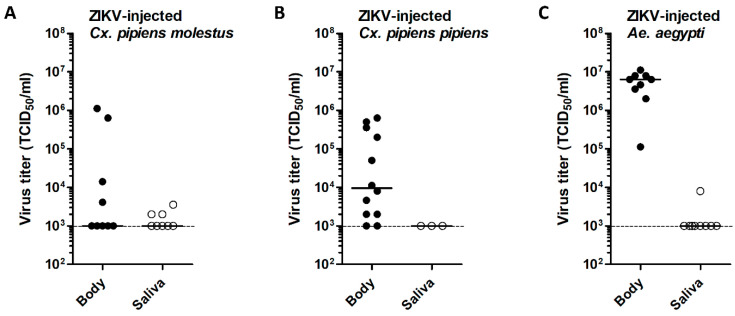
Virus titers in bodies and salivas of *Cx. pipiens molestus*, *Cx. pipiens pipiens*, and *Ae. aegypti* mosquitoes after intrathoracic injection with ZIKV. After injection, mosquitoes were incubated at 28 °C for 14 days. Virus titers of the bodies and salivas were determined by EPDA for (**A**) ZIKV-injected *Cx. pipiens molestus*, (**B**) ZIKV-injected *Cx. pipiens pipiens* (injection dose: 1.0 × 10^4^ TCID_50_), (**C**) ZIKV-injected *Ae. aegypti*. Data points show individual mosquitoes infected with ZIKV. Lines show median virus titers. Dashed lines indicate the detection limit of the EPDA.

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
