# Peer review of "Forced Zika Virus Infection of Culex pipiens Leads to Limited Virus Accumulation in Mosquito Saliva"

_viruses, 2020, doi:10.3390/v12060659_

Round 1

Reviewer 1 Report

ZIKV is one of mosquito-borne viruses, which endangered public health. It has been shown to be carried by mosquito Aedes aegypti. However, the Cx. pipiens mosquitoes are mainly present in temperate regions crossing North-America, Asia and Europe, not Aedes aegypti. To enable ZIKV efficient transmission in these regions, Cx. pipiens would be the ideal carrier. Therefore, the first question for the field needs to be answered is whether ZIKV can infect Cx. pipiens.

Abbo, et al. present a well-organized study to address this important question. They observed ZIKV presence in the saliva of Cx. pipiens upon super infection, which suggested that Cx. Pipiens is an inefficient vector for ZIKV. This is a good news for the public health officials. Moreover, it is in line with a previous report that ZIKV RNA can not be detected in infected Cx. Pipiens mosquitoes (1).

However, in an earlier ZIKV transmission study, it has shown that ZIKV can infect Cx. Pipiens through oral infection (2). It would be valuable for the field if the authors could discuss the potential factors contribute to this discrepancy.

Minor points:

Please provide more detailed description of their statistical analysis. Which software they used to perform Kolmogorov-Smirnov test, a particular R package or a commercially available software?

Ref:

  1. Boccolini D, Toma L, Di Luca M, Severini F, Romi R, Remoli ME, Sabbatucci M, Venturi G, Rezza G, Fortuna C. Experimental investigation of the susceptibility of Italian Culex pipiens mosquitoes to Zika virus infection. Euro Surveill. 2016;21(35). Epub 2016/09/09. doi: 10.2807/1560-7917.ES.2016.21.35.30328. PubMed PMID: 27605056; PMCID: PMC5015456.
  2. Guo XX, Li CX, Deng YQ, Xing D, Liu QM, Wu Q, Sun AJ, Dong YD, Cao WC, Qin CF, Zhao TY. Culex pipiens quinquefasciatus: a potential vector to transmit Zika virus. Emerg Microbes Infect. 2016;5(9):e102. Epub 2016/09/08. doi: 10.1038/emi.2016.102. PubMed PMID: 27599470; PMCID: PMC5113053.

Author Response

> Authors: Guo et al. and also Smartt et al. [1] reported experimental transmission of ZIKV by Culex quinquefasciatus, not Culex pipiens. We have already addressed these studies in the introduction (line 51-52), and the potential factors contributing to the discrepancy between different studies on the vector competence of Culex mosquitoes for ZIKV have been discussed extensively elsewhere [2-4]. Moreover, the discrepancy relates to studies with the southern house mosquito Culex quinquefasciatus whereas we investigated the common house mosquito Culex pipiens, which has, as far as we know, been proven to be an incompetent vector for ZIKV in all published vector competence studies. For these reasons, we chose not to elaborate on the discrepancy for Culex quinquefasciatus in the Discussion of our current paper.

> Authors: The following sentence was added to line 166: ‘All statistical tests were performed using GraphPad Prism 5.’

References

  1. Smartt, C.T.; Shin, D.; Kang, S.; Tabachnick, W.J. Culex quinquefasciatus (Diptera: Culicidae) From Florida Transmitted Zika Virus. Front Microbiol 2018, 9, 768.
  2. Van den Hurk, A.F.; Hall-Mendelin, S.; Jansen, C.C.; Higgs, S. Zika virus and Culex quinquefasciatus mosquitoes: a tenuous link. Lancet Infect Dis 2017, 17, 1014-1016.
  3. Roundy, C.M.; Azar, S.R.; Brault, A.C.; Ebel, G.D.; Failloux, A.B.; Fernandez-Salas, I.; Kitron, U.; Kramer, L.D.; Lourenço-de-Oliveira, R.; Osorio, J.E., et al. Lack of Evidence for Zika Virus Transmission by Culex Emerg Microbes Infect 2017, 6, e90.
  4. Ayres, C.; Guedes, D.; Paiva, M.; Donato, M.; Barbosa, P.; Krokovsky, L.; Rocha, S.; Saraiva, K.; Crespo, M.; Rezende, T., et al. Response to: ‘Lack of evidence for Zika virus transmission by Culex mosquitoes’. Emerg Microbes Infect 2017, 6, e91.

Reviewer 2 Report

This is a well carried out study on the absence of vector competence of Cx pipiens for ZIKV after oral contamination.The methodology and the use of controls validating the study are appropriate. The analyse of the results is strong.

Interestingly, the authors show the virus can infect the salivary glands after intrathoracic inoculation, demontrating the role of digestive barriers in natural conditions.

In the conclusion, I don't understand the sentence (line 317-321) "However, when considering the massive number of mosquitoes in the field and the fact that our laboratory study only measures vector competence and does not take into account all factors contributing to the vectorial capacity of the Cx. pipiens mosquito species [39], we cannot completely rule out the possibility of ZIKV transmission by Cx. pipiens in the field." It sound rather confusing taking into considertion the authors nicely showed the absence of vector competence after oral infection. I suggest to delete this sentence.

Author Response

> Authors: Our study was carried out in the laboratory with different environmental conditions compared to the natural situation in the field. For example, the conditions during larval development (e.g. food source; larval density) in the laboratory are different in the field. Further, we infected mosquitoes using a Hemotek system whereas under natural conditions mosquitoes will feed on infectious hosts. And although we tested very high numbers of mosquitoes, we cannot completely rule out the possibility of ZIKV transmission by Culex pipiens when we consider the massive number of mosquitoes in the field. These and other factors contributing to the vectorial capacity of the mosquito species cannot be covered in a vector competence study, and we therefore think it is valuable to acknowledge this potential caveat in the Discussion.